# Inhibitory Effects of 3-Methylcholanthrene Exposure on Porcine Oocyte Maturation

**DOI:** 10.3390/ijms24065567

**Published:** 2023-03-14

**Authors:** Mengya Zhang, Xin Wang, Qiuchen Liu, Yelian Yan, Changzhi Xu, Zhihua Zhu, Sucheng Wu, Xiaoyu Zuo, Yanfeng Zong, Chunxiao Li, Yunhai Zhang, Zubing Cao

**Affiliations:** Anhui Province Key Laboratory of Local Livestock and Poultry, Genetical Resource Conservation and Breeding, College of Animal Science and Technology, Anhui Agricultural University, Hefei 230036, China

**Keywords:** 3-methylcholanthrene, oocyte maturation, embryo development, pig

## Abstract

3-methylcholanthrene (3-MC) is a highly toxic environmental pollutant that impairs animal health. 3-MC exposure can cause abnormal spermatogenesis and ovarian dysfunction. However, the effects of 3-MC exposure on oocyte maturation and embryo development remain unclear. This study revealed the toxic effects of 3-MC exposure on oocyte maturation and embryo development. 3-MC with different concentrations of 0, 25, 50, and 100 μM was applied for in vitro maturation of porcine oocytes. The results showed that 100 μM 3-MC significantly inhibited cumulus expansion and the first polar body extrusion. The rates of cleavage and blastocyst of embryos derived from 3-MC-exposed oocytes were significantly lower than those in the control group. Additionally, the rates of spindle abnormalities and chromosomal misalignments were higher than those in the control group. Furthermore, 3-MC exposure not only decreased the levels of mitochondria, cortical granules (CGs), and acetylated α-Tubulin, but also increased the levels of reactive oxygen species (ROS), DNA damage, and apoptosis. The expression of cumulus expansion and apoptosis-related genes was abnormal in 3-MC-exposed oocytes. In conclusion, 3-MC exposure disrupted the nuclear and cytoplasmic maturation of porcine oocytes through oxidative stress.

## 1. Introduction

3-methylcholanthrene (3-MC) is known as a dioxin-like compound and is also considered as an environmental pollutant with dangerous carcinogenic properties [1,2]. 3-MC is released into the environment because of the inadequate combustion of macromolecular organic compounds derived from trees, tobacco, coal, gasoline, waste incineration, straw incineration, coal furnace combustion, and chemical products. 

Dioxin-like compounds are mainly present in water, soil, and air. These pollutants are usually absorbed and accumulated in the body via the food chain and cause harmful effects on the body, such as immune toxicity [3,4,5], reproductive toxicity [6,7,8,9], and developmental toxicity [10,11,12]. In a recent study, a certain amount of dioxin was discovered in human blood, breast milk, and normal cystic liquid [13,14,15] and dioxin exposure hindered early embryo development and embryo implantation. During early embryonic development in macaques, dioxin exposure can cause embryo loss and death [16]. Dioxins reduce the hatchability of chicken embryos and increase the death rate after hatching [8]. 

Dioxins inhibited early embryo cleavage in mice, and mouse embryos were sensitive to dioxins in a concentration-dependent manner [6,7]. The offspring of dioxin-exposed rats showed decreased reproductive ability and a range of developmental disorders; the number of primordial follicles in female mice was reduced, apoptosis pathways were activated, and embryonic mortality increased. Dioxins could cause miscarriages and premature births in humans, and even birth defects or stunted growth in fetuses [9]. Hence, dioxin exposure causes ovarian dysfunction in female animals, inhibits estrogen secretion, and even leads to miscarriage and infertility [17,18]. For the male reproductive system, dioxin reduces the number and maturity of sperm cells, causing maternalism, which significantly affects reproduction in animals [19]. Studies have shown that exposure to 3-MC causes damage to cells, tissue, and organs [20,21,22,23,24]. Numerous studies have shown that 3-MC exposure rechanges cell adhesion and induces cell apoptosis [25,26,27], affecting the immunity and metabolism of tissue, which in turn leads to the retention of carcinogenic substances in tissues and makes organs cancerous [28,29,30,31]. Oocyte maturation is a key physiological event and critical for female animal reproduction, but the toxic effects of 3MC on oocyte maturation have been not reported. Therefore, we hypothesize that 3-MC exposure may impair oocyte maturation and development.

In current study, porcine oocytes were used as a research model to study the effect of 3-MC exposure on oocyte meiotic maturation because pigs have many similarities with humans in terms of body structure, organ size, physiological function, and pathological characteristics [32,33,34]. The results provide some clues for developing effective strategies to mitigate the toxic effects of 3-MC for animal and human oocytes. 

## 2. Results

### 2.1. 3-MC Exposure Inhibited Porcine Oocyte Meiotic Maturation and Early Embryonic Development

To determine the effect of 3-MC exposure on the meiotic maturation of porcine oocytes, 3-MC of different concentrations (0 μM, 25 μM, 50 μM, 100 μM) was added into the maturation medium. Exposure to different concentrations of 3-MC inhibited the expansion of cumulus cells and significantly reduced the rate of pb1 extrusion (Figure 1A–C) (*p* < 0.05). The concentration of 100 μM not only inhibited oocyte meiosis but also allowed a part of the oocytes to be matured for other investigations. Hence, 100 μM 3-MC was selected for the subsequent studies. Next, the oocytes with pb1 were parthenogenetically activated and developed to the blastocyst stage. The cleavage and blastocyst rates for the 3-MC-exposed group were significantly lower than those of the control group (Figure 1D–F) (*p* < 0.05). In addition, RT-qPCR results showed that 3-MC exposure significantly reduced the expression of *PTX3* and *PIGS2* (*p* < 0.05) and increased the expression of *TNFAIP6*, but did not affect the expression of *CD44* and *HAS2* (Figure 1G). 

### 2.2. 3-MC Exposure Disrupted the Spindle Assembly and Chromosome Alignment in Oocytes

To evaluate whether 3-MC exposure impaired spindle assembly and chromosome alignment, anti-α-Tubulin antibody staining was performed on MII-stage oocytes exposed to 3-MC. The results showed that the spindle of the control oocytes was normally located between the poles and had the standard fusiform with the chromosome liner and were neatly arranged on either side of the equator. However, the oocytes in the 3-MC exposed group with mass spindles showed irregular shapes and chromosome arrangement disorders (Figure 2A). Statistical analysis showed that 3-MC exposure significantly increased the rate of spindle abnormalities and chromosome misarrangement (Figure 2B,C) (*p* < 0.05). These results indicated that 3-MC exposure caused cytoskeleton defects in oocytes. 

### 2.3. 3-MC Exposure Inhibited the Acetylation of α-Tubulin in Oocytes

To test whether 3-MC exposure affected acetylated α-tubulin levels in oocytes, oocytes exposed to 3-MC were stained with anti-acetylated α-tubulin. As shown in Figure 3A, signals of acetylated tubulin in 3-MC exposed oocytes almost disappeared. Additionally, fluorescence intensity analysis revealed that the acetylated α-tubulin levels of oocytes exposed to 3-MC significantly decreased compared with the control group (Figure 3B) (*p* < 0.05). This data suggested that 3-MC exposure diminished the levels of acetylated α-tubulin in oocytes.

### 2.4. 3-MC Exposure Disrupted the Polymerization of Actin Filaments in Oocytes

To investigate the effect of 3-MC exposure on actin polymerization, oocytes were stained with Anti-F-actin antibody. We observed that actin filaments in the control group were well and uniformly distributed on the cytoplasmic membrane and accompanied by a strong fluorescent signal. In contrast, oocytes in the 3-MC exposure group showed impaired actin filament assembly with uneven and discontinuous distribution at the plasma membrane (Figure 4A,B) (*p* < 0.05). Fluorescence intensity analysis showed that the signaling intensity of F-actin in 3-MC-exposed oocytes significantly decreased compared with that in the control group (Figure 4C) (*p* < 0.05). The results suggested that 3-MC exposure impaired the actin cytoskeleton in oocytes.

### 2.5. 3-MC Exposure Disrupted the Distribution of Mitochondria and Cortical Granules in Oocytes

The mitochondrion is an important indicator for cytoplasmic maturation. MitoTracker Red CMX Ros staining was performed to analyze the mitochondrial distribution. The results showed that mitochondria in the control group were evenly distributed around the plasma membrane and accompanied by a strong fluorescence signal, while mitochondria in the 3-MC-exposed oocytes were not evenly distributed, with a trend of normal distribution, and fluorescence signals were significantly reduced (Figure 5A,B) (*p* < 0.05). To determine the effect of 3-MC exposure on CG distribution, oocytes were stained with PNA and their fluorescence intensity was analyzed. The CGs of oocytes in the control group were distributed largely and uniformly in the cortical area with a strong fluorescence signal. In contrast, the CGs of oocytes in the 3-MC exposure group were distributed unevenly in the cortical area with a weak fluorescence signal (Figure 5C,D) (*p* < 0.05). The results suggested that 3-MC affected the migration and distribution of mitochondria and CGs in oocytes. 

### 2.6. 3-MC Exposure Increased the Levels of DNA Damage and Apoptosis in Oocytes

3-MC exposure can cause different degrees of oxidative stress in various tissues and cells. To explore whether 3-MC exposure caused DNA damage in porcine oocytes, DCFH-DA staining was performed. The results showed that the fluorescence intensity of the 3-MC exposed oocytes was higher in comparison with the control group, indicating that 3-MC exposure increased the ROS levels of oocytes (Figure 6A,B) (*p* < 0.05). Then, anti-γ.H2AX body and Annexin-V staining were used to detect the level of DNA damage and apoptosis of oocytes and the fluorescence intensity was analyzed. Data analysis showed that the fluorescence intensity of the γ.H2AX and Annexin-V in the 3-MC exposure group was significantly higher than that in the control group (Figure 6C–F) (*p* < 0.05). RT-qPCR results revealed that the expressions of *CASPESE3* and *BCL2* genes were downregulated (Figure 6G) (*p* < 0.05). These results indicated that 3-MC exposure could significantly increase the level of DNA damage and cell apoptosis.

## 3. Discussion

With increasingly serious environmental pollution, environmental pollutants have caused more and more harm to organisms. The concerning impact of environmental pollutants on animal reproduction and development has been widely investigated [1,2]. 3-MC is a common environmental pollutant with strong toxicity and carcinogenicity; exposure to 3-MC can lead to various cancers and tumor diseases [3,4]. Additionally, 3-MC is an ovotoxin that can affect oocyte growth, and partial studies have revealed the effects of 3-MC on the reproductive system in small model organisms [17,18,19]. However, the toxic effects and mechanisms of 3-MC on mammalian oocyte maturation are not known. In model animals such as zebrafish, mice, and rats, 3-MC exposure causes tissue and cell lesions. Consistent with these results, we found that 3-MC exposure inhibited the nuclear and cytoplasmic maturation of porcine oocytes and affected early embryonic development.

Previous studies have proven that cumulus diffusion was related to oocyte maturation and affected embryo development [35,36]. In this study, we found that 3-MC exposure inhibited the first polar body expulsion and cumulus expansion. Dioxins are known to cause deformities and even death in mammalian embryos [37,38,39,40]. Similar to these data, our results revealed that 3-MC exposure reduced cleavage and blastocyst rate. The transcriptions of *CD44*, *HAS2*, *PTGS2*, *TNFAIP6* and *PTX3* were expressed in cumulus granulosa cells and played an important role in cumulus diffusion [41,42] and the expressions of *PTX3* and *PIGS2* were downregulated, while the expression of *TNFAIP6* was upregulated. Likely, *TNFAIP6* was not dominant in porcine cumulus expansion, so its upregulation did not alleviate the inhibition of cumulus expansion. These results indicated that 3-MC exposure interfered with the expression of related genes, inhibited cumulus expansion, led to the arrest of oocyte maturation, accelerated oocyte apoptosis, and decreased embryo development efficiency.

We further explored the specific mechanism underlying 3-MC’s toxic effects on oocyte maturation. Nuclear and cytoplasmic maturation are two important indicators of oocyte maturation [32,33,34]. We detected the important indicators related to nuclear maturation. Microtubules were the main components of the cytoskeleton [43], having the kinetic characteristics of polymerization and depolymerization, and played a key role in the maintenance of cell morphology, cell division, and material transport. The spindle was composed of microtubules. The arrest of oocyte meiosis is usually associated with the damage of cytoskeleton structure and abnormal spindle assembly, and chromosome alignment can cause meiotic errors and inhibit oocyte maturation [32,33,34,43]. The spindle played an important role in the movement and distribution of chromosomes and the generation of polar bodies during oocyte meiosis. We detected that 3-MC exposure disrupted the structure of the spindle, resulting in chromosome dislocation. Tubulin acetylation can protect the structure of microtubules, and changes in the level of acetylated microtubules can lead to spindle structural defects [44,45,46,47]. We further observed that the acetylation level of α-Tubulin significantly decreased. Actin was another important component of the cytoskeleton and is involved in intracellular vesicle and organelle motility. Actin filaments played a key role in the process of cytogenesis such as asymmetric positioning of the spindle, chromosome polymerization, chromosome separation and cytokinesis. The migration of the spindle to the oocyte cortex was completed in an actin-dependent manner [48,49,50,51,52]. We found that 3-MC exposure decreased the acetylation level of α-tubulin and affected the normal assembly of actin. Considering the analyses for the abovementioned indicators, we speculated that 3-MC exposure reduced the level of tubulin acetylation in oocytes, affected the stability of microtubules, and interfered with the polymerization of actin, thus affecting the assembly, migration and localization of the spindle, and finally led to the impaired nuclear maturation of oocytes.

Next, we examined the indicators related to cytoplasmic maturation. The mitochondrion is an important organelle and the energy factory of eukaryotic cells. Both spindle formation and chromosome movement depend on the expression and activity of motor proteins that use ATP as an energy source. During oocyte maturation, the distribution of mitochondria changes dynamically, being an important indicator of cytoplasmic maturation [49,53,54]. We found in this study that 3-MC exposure perturbed mitochondria distribution and disrupted mitochondrial integrity. CGs were synthesized from Golgi enrichment, along with oocyte maturation and migration to the subcortical area. The secretion of CGs and their binding with the plasma membrane are essential prerequisites for blocking polyspermy into oocytes [2,3,6,7]. The uniform distribution of CGs in the subcortical area was an important symbol of oocyte cytoplasmic maturation and affected the subsequent embryonic development. Thus, it is possible that the low developmental competence of 3-MC-exposed oocytes could be due to the abnormal distribution of mitochondria and CGs.

Oxidative stress is a basic self-protection mechanism of living organisms. When cells are subjected to harmful stimuli, the oxidation–antioxidant system is unbalanced and causes a large number of reactive oxygen species (ROS) radicals to be produced, causing cell apoptosis [55,56]. Meanwhile, ROS can directly or indirectly oxidize/damage DNA [11,18,32]. Exposure to 3-MC disrupted the normal growth environment of oocytes. We found that the levels of ROS, DNA damage, and apoptosis in porcine oocytes exposed to 3-MC were all significantly increased. ROS is an important intracellular second messenger that is essential for the regulation of ovulation, meiosis, and embryonic development in female animals and for maintaining fertilization [57,58]. The accumulation of ROS can destroy the structure of the spindle and the distribution of mitochondrial membrane potential, leading to oocyte senescence and delayed embryonic development [59]. We speculated that the failure of porcine oocyte maturation might be related to increased ROS levels. *BCL2* is an important regulator of anti-apoptotic cells, while *BAX* and *CASPASE3* are pro-apoptotic proteins. The ratio of expression levels of these two genes can determine whether cells are proliferating or apoptotic [60]. The abnormal expression of *CASPASE3* and *BCL2* might be related to the apoptosis of oocytes. 

In conclusion, these results demonstrated that 3-MC exposure impaired the maturation of porcine oocytes through oxidative stress. This study elucidated the effect of 3-MC on porcine oocyte maturation and its potential mechanism of action, and further proved the reproductive toxicity of 3-MC to mammals. It could provide new hints for developing some strategies to mitigate the toxic effects of 3-MC in agricultural animals and humans. 

## 4. Materials and Methods

### 4.1. Experimental Design

Pigs were taken as research materials to explore the effect of 3-MC on pig oocyte maturation meiosis. Oocytes were randomly divided into four groups, including the control group, 25 μM 3-MC exposure group, 50 μM 3-MC exposure group, and 100 μM 3-MC exposure group. Through oocyte maturation culture in vitro, it was concluded that the concentration of 100 μM 3-MC had the most significant effect on the rate of the first polar body extrusion (control group: 74.63% ± 0.031; 100 μM 3-MC exposure group: 51.53% ± 0.012) and the expansion of the cumulus (control group: 0.2043 ± 0.028; 100 μM 3-MC exposure group: 0.1040 ± 0.019). Therefore, subsequent experiments were carried out on the basis of 100 μM concentrations.

Experiment 1 was conducted to determine the effect of 3-MC exposure on porcine oocyte maturation and embryo development. We detected the expression of genes related to cumulus expansion. The oocytes with the first polar body were activated by parthenogenesis, and embryos were cultured in 38.5 °C, 5% CO_2_ incubator under saturated humidity; then, the cleavage and blastocyst rates were calculated to evaluate the effect of 3-MC exposure on oocyte meiosis maturation and embryo development.

Experiment 2 was conducted to determine the effect of 3-MC exposure on porcine oocyte nuclear maturation. Chromosome arrangement, spindle morphology, acetylated α-tubulin levels, and actin dynamics were evaluated in the control and 3-MC exposure groups.

Experiment 3 was conducted to determine the effect of 3-MC exposure on porcine oocyte cytoplasmic maturation. Cortical granule distribution and mitochondrial integrity were determined in the control and 3-MC exposure groups.

Experiment 4 was conducted to determine the effect of 3-MC exposure on the levels of reactive oxygen species. ROS, DNA damage, apoptosis levels, and the expressions of apoptosis-related genes were detected.

The reagents used in this study were purchased from Sigma (Sigma-Aldrich, St Louis, MO, USA) unless otherwise stated. All animal experiments were conducted according to the guidelines of the Institutional Animal Care and Use Committee (IACUC) of Anhui Agricultural University.

### 4.2. Preparation of 3-MC

3-MC (SUPELCO, 442388) was dissolved in DMSO to obtain 100 mM stock solution, and then the required working concentration (25 μM, 50 μM, 100 μM) with maturation medium. 3-MC of 100 μM was used for the subsequent experiments.

### 4.3. Oocyte In Vitro Maturation

Ovaries were collected from a local slaughterhouse. Follicular fluid was aspirated from antral follicles at 3–6 mm in diameter. Cumulus–oocyte complexes (COCs) were selected with two or more layers of compact cumulus cells under a stereomicroscope. Subsequently, COCs were cultured in four plates containing 400 μL vitro maturation medium (Medium 199 supplemented with 10% porcine large follicular fluid, 5% FBS, 10 IU/mL PMSG, 5 IU/mL hCG, 100 ng/mL L-Cysteine, 10 ng/mL EGF, 0.23 ng/mL melatonin, 2.03 × 10^−5^ ng/mL LIF, 2 × 10^−5^ ng/mL IGF-1, 4 × 10^−5^ ng/mL FGF2, 100 U/mL penicillin, and 100 mg/mL streptomycin) for 44 h at 38.5 °C, 5% CO_2_, and saturated humidity. COCs at different stages of meiosis were incubated in Ca^2+^ and Mg^2+^ free DPBS (Gibco, Grand Isle, NY, USA) supplemented with 1 mg/mL hyaluronidase to remove the cumulus cells. Oocytes with the completion of nuclear maturation were confirmed by the presence of the first polar body (pb1). The medium of the control group was supplemented with DMSO in equal proportion to the experimental group to eliminate the toxicity of DMSO. Because of the toxicity of DMSO itself, the dosage of DMSO in the medium was not more than 1‰. Each group consisted of 50 oocytes.

### 4.4. Parthenogenetic Activation and Embryo Culture

The oocytes with pb1 were washed twice quickly with activation medium (0.1 mM CaCl_2_, 1 mM MgCl_2_, and 0.01% polyvinyl alcohol added to 0.3 M mannitol). Then, oocytes were moved to a 1 mm fusion tank containing 70 μL activation medium and stimulated with two pulses of direct current (1.56 KV/cm, 80 ms). The electrically activated oocytes were then washed twice with PZM-3 (Weigh 0.6312 g NaCl, 0.005 g MgSO_4_, 0.2106 g NaHCO_3_, 0.0746 g KH_2_PO_4_, 0.05 g BSA, 2 mL EAA, 1 mL NEAA, 0.05 g Myo-Inositol, 0.0022 g Sodium pyruvate, 0.0616 g Ca-lactate·5H_2_O, 0.0146 g L-Glutamine, 0.0546 g Hypotaurine, and 1 mL of imported bispecific antibody dissolved in 50 mL embryo culture medium, fully dissolved, and then fixed to 100 mL with embryo culture water), and twice with PCC (PZM-3 supplemented with 10 mg/mL cycloheximide and 10 mg/mL cytochalasin B). Then, the oocytes were transferred to medium and cultured under 38.5 °C, 5% CO_2_, and saturated humidity for seven days, and observed and measured at all stages of embryonic development. 

### 4.5. Real-Time Quantitative PCR (RT-qPCR)

Total RNA was extracted from groups of ten oocytes using RNeasy Mini Kit (QIAGEN, 74034). The RNA was quantified using a NanoDrop 2000 instrument (Thermo Scientific, Waltham, MA, USA) and was converted to cDNA using a Reverse Transcription Kit (biosharp, BL696A). Real-time PCR was carried out using Universal SYBR qPCR Master Mix (biosharp, BL697A) and a StepOne Plus machine (Applied Biosystems, New York, NY, USA). One cycle was performed for predenaturation at 95 °C for 2 min, 40 cycles for denaturation at 95 °C for 15 s, and annealing at 60 °C for 30 s. Three biological replicates were conducted for each gene. The primers used in this study are listed in Appendix A.

### 4.6. Immunofluorescence (IF) Staining

Oocyte cells were fixed in a 4% paraformaldehyde (PFA) solution for 15 min, permeabilized with 0.5% Triton X-100 (in DPBS) for 30 min at room temperature (RT), and then blocked with 2% BSA (in DPBS) at RT for 1 h to suppress the non-specific binding of IgG. Samples were incubated in a blocking solution containing primary antibodies (primary antibodies were diluted with 2%BSA according to the dilution ratio provided in Appendix A) against target proteins overnight at 4 °C. After washing four times with DPBS + 0.3%PVP, 15 min each time, samples were incubated for 1 h in blocking solution containing secondary antibodies (secondary antibodies were diluted with 2%BSA in the ratio of 1 to 200) in the dark at 37 °C, followed by washing three times with DPBS + 0.3% PVP, 15 min/each time. Finally, 4, 6-diamidino-2-phenylindole dihydrochloride (DAPI) was used to stain the nuclei for 10 min at room temperature. Samples were placed on glass slides with coverslips and examined with a confocal laser-scanning microscope (Olympus, Tokyo, Japan). Then fluorescence intensity analysis was performed for each group consisting of 20~30 oocytes. The detailed information regarding primary and secondary antibodies used is listed in Appendix A.

### 4.7. Determination of Mitochondrial Distribution

The MII oocytes were transferred to MitoTracker Red CMXRos. MitoTracker Red CMXRos (Thermo Fisher, Waltman, MA, USA) and diluted with medium at a dilution ratio of 1 to 2000. Oocytes were incubated in a 38.5 °C, 5% CO_2_ incubator for 30 min, then washed 3 times with DPBS + 0.3% PVP for 5 min each time. Observation was performed with a confocal microscope (Olympus, Tokyo, Japan). Each group consisted of 20~30 oocytes.

### 4.8. DCFH-DA Staining

MII oocytes were transferred to the DCFH-DA (Beyotime, Nantong, China), and the working concentration of DCFH-DA was 10 μM. Oocytes were incubated in a 38.5 °C, 5% CO_2_ incubator for 30 min, then washed 3 times with DPBS + 0.3% PVP for 5 min each time. Observation was performed with a confocal microscope (Olympus, Tokyo, Japan). Each group consisted of 20~30 oocytes.

### 4.9. Annexin-V Staining

MII oocytes were transferred to Annexin-V (Vazyme, Nanjing, China). The Annexin-V was diluted in the dilution ratio of 1 to 9 by the binding buffer. Oocytes were incubated in a 38.5 °C, 5% CO_2_ incubator for 30 min, then washed 3 times with DPBS + 0.3% PVP for 5 min each time. Observation was performed with a confocal microscope (Olympus, Tokyo, Japan). Each group consisted of 20~30 oocytes.

### 4.10. Calculation of Expansion Area of Cumulus Cells

Oocytes with the same degree of cumulus cell encapsulation matured for 44 h in vitro and were then photographed under an inverted microscope. The diffused area of cumulus was measured using a measuring tool in the software Image-Pro Express 6.0 and calculated according to the previously published formula, namely, area = length × width × 0.7854. The distance between the two points with the largest diffusion distance was taken as the length, and the distance between the two points with the nearest diffusion distance was taken as the width. Each group consisted of 15~25 oocytes.

### 4.11. Statistical Analysis of Data

SPSS 26.0 statistical software was used for data analysis. All experiments were carried out at least three times and the data were analyzed by one-way analysis of variance (ANOVA) or independent sample t test. The data were presented as mean ± standard error of the mean (mean ± S.E.M). *p* value < 0.05 indicates a statistically significant difference.

## 5. Conclusions

Taken together, 3-MC exposure disrupted the cytoskeleton, significantly increased the levels of reactive oxygen species, and caused the aberrant expression of cumulus expansion and apoptosis-associated genes. These results suggest that 3-MC exposure caused abnormal meiotic maturation of porcine oocytes by affecting nuclear and cytoplasmic maturation. Further studies to search for effective strategies to reduce the toxicity of 3-MC in porcine oocytes are needed.

## Figures and Tables

**Figure 1 ijms-24-05567-f001:**
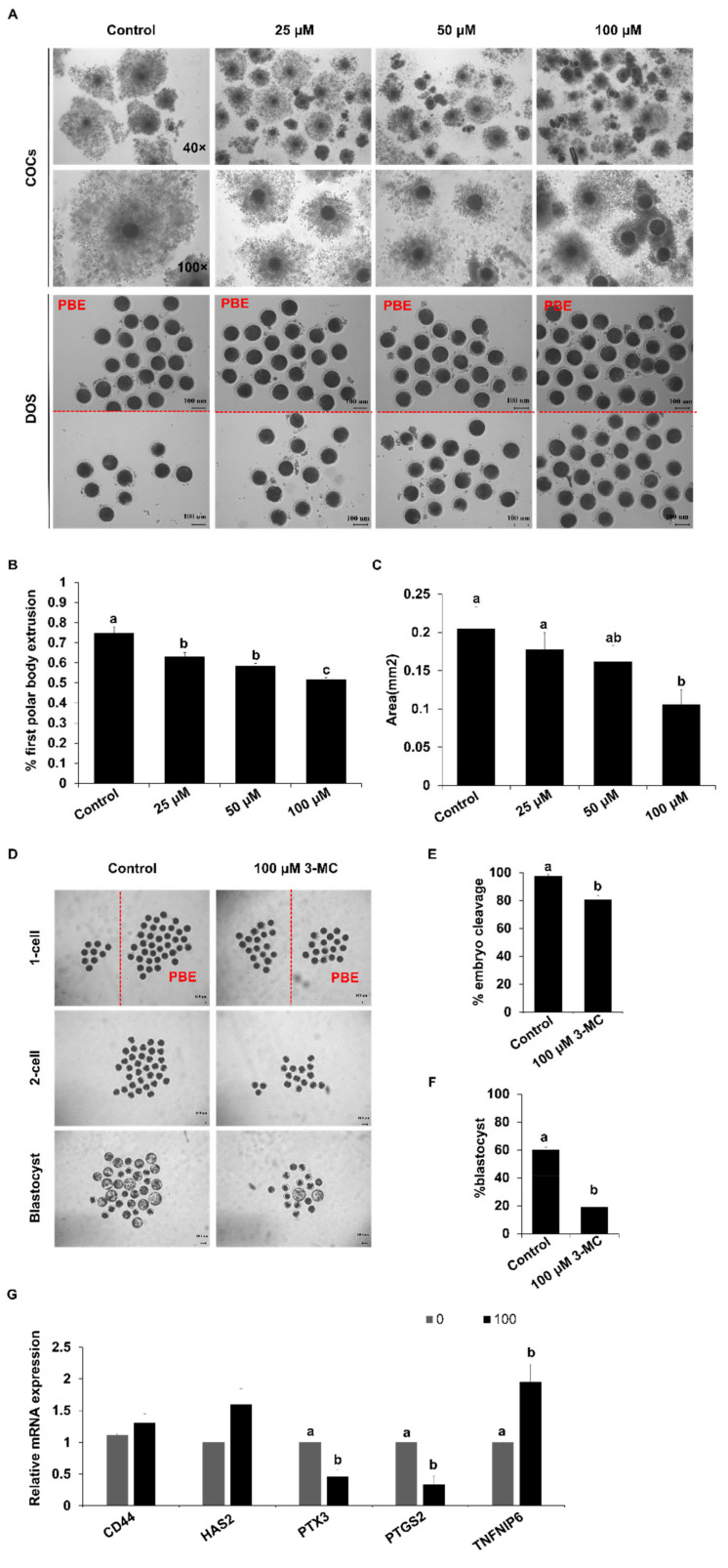
The effects of 3-MC exposure on porcine oocyte maturation and embryonic development. (**A**) Representative images of COCs and denuded oocytes before and after maturation. COCs were maturated for 44 h in vitro in control and 3-MC treatment groups, scale bar 100 μM; (**B**) the rate of the PBE for oocytes in control and 3-MC exposure groups; (**C**) cumulus expansion area for COCs in control and 3-MC exposure groups; (**D**) representative images of parthenogenetic embryos derived from oocytes with pb1 extrusion in control and 3-MC exposure groups, scale bar 100 μM; (**E**,**F**) the rates of two-cell embryos and blastocysts in control and 3-MC exposure groups; (**G**) expression of cumulus expansion-related genes in oocytes from each group. All data were shown as mean ± S.E.M and different letters on the bars indicate significant differences. (*p* < 0.05).

**Figure 2 ijms-24-05567-f002:**
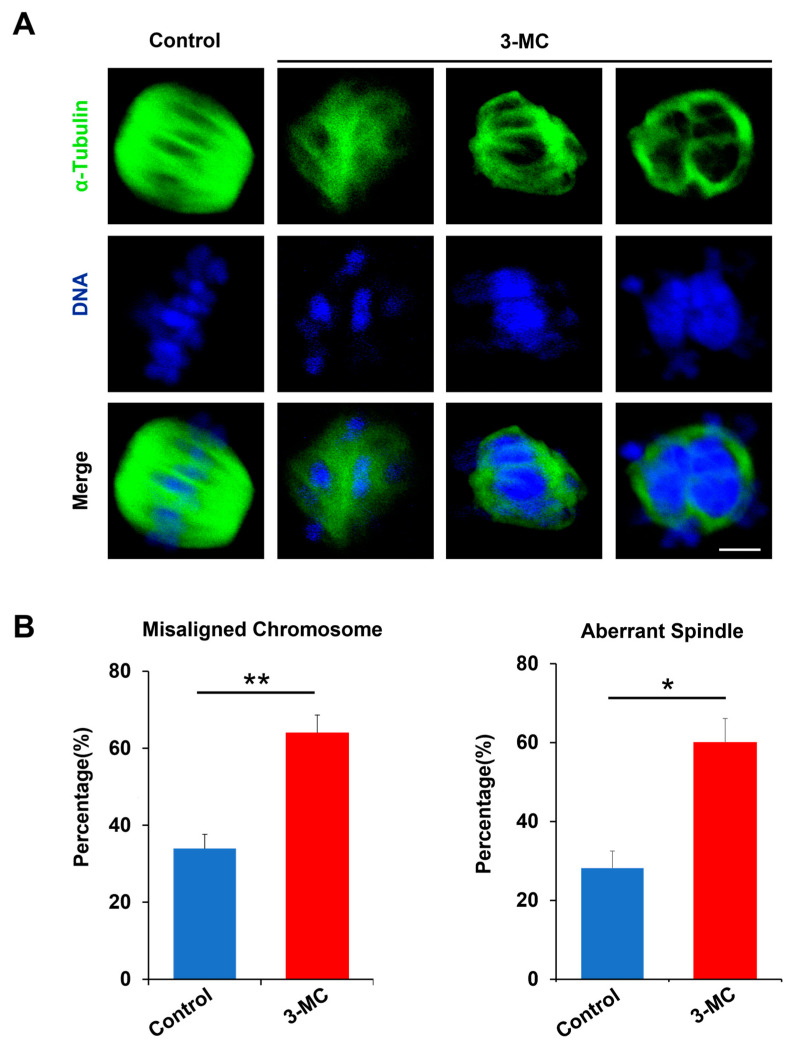
Effects of 3-MC exposure on the chromosome alignment and spindle assembly in porcine oocytes. (**A**) Representative images of chromosome and spindle morphologies in control and 100 μM 3-MC treatment oocytes were stained with anti-α-Tubulin antibody (green) to visualize the spindles and were counterstained with 4′,6-diamiolino-2-pnenylindole (bule) to visualize the chromosomes, scale bar 3.5 μM; (**B**) the rate of misaligned chromosomes and aberrant spindles was recorded in each group. All the data were shown as mean ± S.E.M of at least three independent experiments. ** *p* < 0.01, * *p* < 0.05.

**Figure 3 ijms-24-05567-f003:**
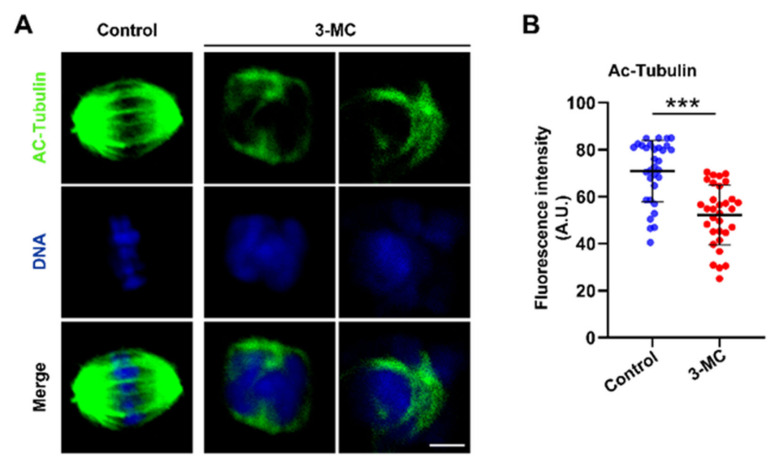
Effects of 3-MC exposure on the acetylation level of α-tubulin in porcine oocytes. (**A**) Representative images of acetylated α-tubulin in the control and 100 μM 3-MC exposure groups, scale bar 3.5 μM; (**B**) the fluorescence intensity of acetylated α-tubulin was quantitatively analyzed in each group. Data were presented as mean ± S.E.M of at least three independent experiments. *** *p* < 0.001.

**Figure 4 ijms-24-05567-f004:**
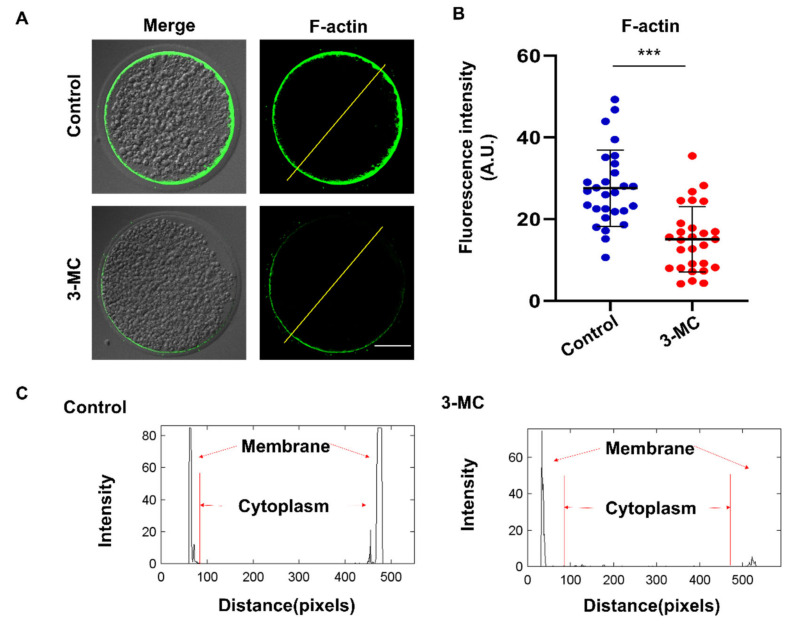
Effects of 3-MC exposure on the actin polymerization in porcine oocytes. (**A**) Representative images of actin filaments in control and in the 100 μM 3-MC treatment group, scale bar 25 μM; (**B**) the fluorescence intensity of actin signals was quantitatively analyzed in each group. (**C**) The fluorescence intensity profiling of actin filaments in each group. Lines were drawn through the oocytes, and pixel intensities were quantified along the lines; data were presented as mean ± S.E.M of at least three independent experiments. *** *p* < 0.001.

**Figure 5 ijms-24-05567-f005:**
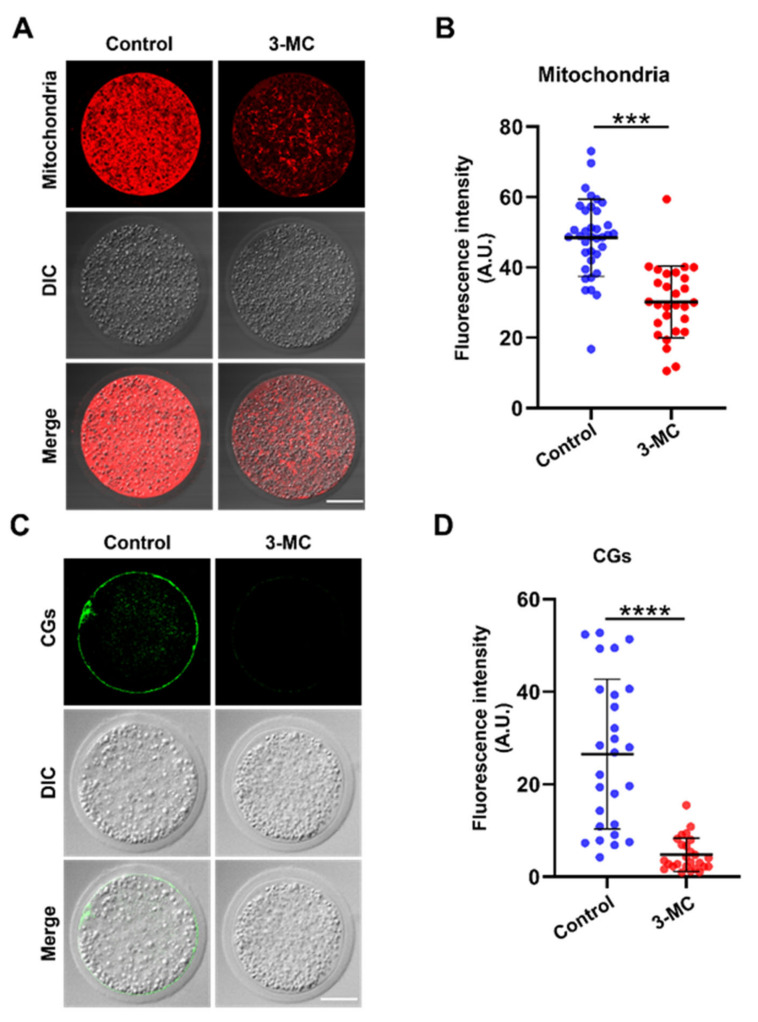
Effects of 3-MC exposure on the distribution of mitochondria and CGs in porcine oocytes. (**A**) Representative images of mitochondria staining in the control and 3-MC treatment groups, scale bar 25 μM; (**B**) the fluorescence intensity of mitochondria signals was quantitatively analyzed in each group. Data were presented as mean ± S.E.M of at least three independent experiments. *** *p* < 0.001; (**C**) representative images of CG localization in the control and 3-MC treatment oocytes, scale bar 25 μM; (**D**) the fluorescence intensity of CG signals was measured in each group. Data were presented as mean ± S.E.M of at least three independent experiments. **** *p* < 0.0001.

**Figure 6 ijms-24-05567-f006:**
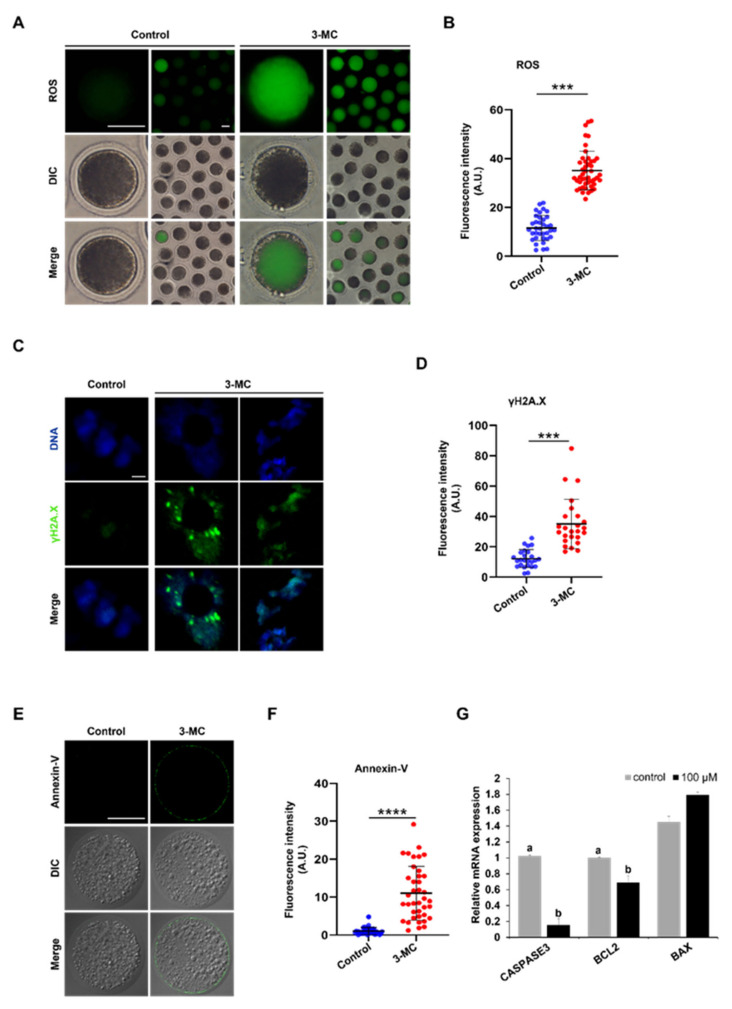
Effects of 3-MC exposure on the ROS, DNA damage, and apoptosis levels in porcine oocytes. (**A**,**E**) Representative images of DCFH-DA and Annexin-v staining in the control and 3-MC treatment oocytes, scale bar 50 μM; (**C**) representative images of γ.H2AX staining in each group, scale bar 2.5 μM; (**B**,**D**,**F**) the fluorescence intensity of ROS, DNA damage, and apoptosis level were measured in control and 3-MC treatment oocytes. All the data were presented as mean ± S.E.M of at least three independent experiments. *** *p* < 0.001, **** *p* < 0.0001; (**G**) expression of apoptosis-related genes in the control and 3-MC treatment groups. All data were shown as mean ± S.E.M and different letters on the bars indicate significant differences (*p* < 0.05).

## Data Availability

The data presented in this study are available on request from the corresponding author.

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
