# Peer review of "Inhibitory Effects of 3-Methylcholanthrene Exposure on Porcine Oocyte Maturation"

_ijms, 2023, doi:10.3390/ijms24065567_

Round 1
Reviewer 1 Report
This study showed the toxic effects of 3-MC exposure on oocyte maturation and embryo development. 100 μM 3-MC significantly inhibited cumulus expansion and oocyte maturations and apoptosis-related genes was abnormal in 3-MC-exposed oocytes. Authors conclude that 3-MC exposure disrupts nuclear and cytoplasmic maturation of porcine oocytes through oxidative stress. However, this manuscript should improve before publications
Study design
Concentrations of MC used in the current study is very high, far from physiologically exposure levels. Authors should state the reason why the high doses was used here.
Experimental design
“Pig organs are closest to human organs in physiological function, shape and size, which is the best donor for human organ transplantation and provides a model for human disease research.” this sentence should be deleted because here is the section of Experimental design.
Author Response
Point 1: Concentrations of MC used in the current study is very high, far from physiologically exposure levels. Authors should state the reason why the high doses was used here.
Response 1: Thank you for helpful comments from the reviewer. We are careful about the choice of 3-MC concentration. Through literature investigation, with reference to the concentration of 3-MC on other species, we set the concentration gradient of 0 µM, 0.05 µM, 0.5 µM, 5 µM and 50 µM, but the effect on pig oocyte maturation and cumulus expansion was not significant. So on this basis, the concentration gradients were reset to 0 µM, 25 µM, 50 µM and 100 µM. 100 μM 3-MC significantly inhibited cumulus expansion and the first polar body extrusion. And the mortality rate of oocytes is maintained at a low level. Therefore, subsequent experiments were carried out based on 100 μM. In addition, 3-MC is a dioxin compound,in the working environment of occupational groups such as waste incinerators, chemical producers and metal production personnel, the exposure level of dioxins is much higher than the physiological exposure level, and the dioxin content detected in their blood is 40 times that of normal people. Once dioxins enter the body, they remain in the body for a long time because they are chemically stable and highly fat-soluble, and accumulate in the body for a long time. Their half-life in the body is estimated at 7 to 11 years. In the environment, dioxins tend to accumulate in the food chain. The higher the animal's position in the food chain, the higher the level of dioxin accumulation. So I think the 100 concentration used in this experiment is reasonable and convincing.
Point 2: “Pig organs are closest to human organs in physiological function, shape and size, which is the best donor for human organ transplantation and provides a model for human disease research.” this sentence should be deleted because here is the section of Experimental design.
Response 2: We are grateful for the reviewer’s good comments. We have deleted these sentences in L278-280 in the section of Experimental design in our revised manuscript.

Reviewer 2 Report
The study reports interesting, detailed data on 3-methylcholanthrene effect on porcine oocytes. There are no concerns about design and description. Only small concerns are listed below.
Sometimes English requires correction.
What is 3-methylcholanthrene exposure dose to a living organism? Give its half-life and information on how it is metabolized.
Lines 60-66- the clear aim of the study should be provided, not results
Microphotographic documentation is too small
Lines 269-270 are repeated of earlier information
Lines274-276 - provide confirmation (pilot data or/and references)
Author Response
Point 1: Sometimes English requires correction. What is 3-methylcholanthrene exposure dose to a living organism? Give its half-life and information on how it is metabolized.
Response 1: We are grateful for the reviewer’s good comments. 3-MC is a dioxin compound, normal people through the respiratory route exposure of dioxin content is about 0.03pgTEQ/L, but in some special cases through respiratory exposure of dioxin content is much higher than physiological level, some surveys show that the blood of waste incineration practitioners in the dioxin content is 806pgTEQ/L, which is about 40 times the level of normal people. The biological half-life of dioxins is long, 2,3,7,8 TCDD is 10~15 days in mice, 12~31 days in rats, and up to 5~10 years in humans (average 7 years). Dioxins cannot be detoxified by metabolism, and after leaving the cells, they will circulate to the liver with the blood, and then directly excrete into the intestine through bile, and then most of them return from the intestine through free diffusion, and then enter the liver through the portal vein, and then excrete from the bile, and cycle.
Point 2: Lines 60-66- the clear aim of the study should be provided, not results
Response 2: We are grateful for the reviewer’s good suggestions. We have revised these sentences in L63-67 in the revised manuscript.
Point 3: Microphotographic documentation is too small
Response 3: Thank you for the good reminder. All microphotographic documentation have been resized in the revised manuscript.
Point 4: Lines 269-270 are repeated of earlier information
Response 4: Thank you for the good comment. We have deleted these sentences in L278-280 in the section of Experimental design in our revised manuscript.
Point 5: Lines274-276 - provide confirmation (pilot data or/and references)
Response 5: We are grateful for the good comment. We have supplemented the pilot data in L285-288 in the revised manuscript.
